# Land surface albedo and vegetation feedbacks enhanced the Millennium drought in south-east Australia

Jason P. Evans[1], Xianhong Meng[2], Matthew F. McCabe[3]

[1]Climate Change Research Centre and ARC Centre of Excellence for Climate System Science, University of New South Wales, Sydney, 2052, Australia
[2]Key Laboratory of Land Surface Process and Climate Change in Cold and Arid Regions, Northwest Institute of Eco-Environment and Resources, Chinese Academy of Science, Lanzhou, China
[3]Water Desalination and Reuse Center, Division of Biological and Environmental Sciences and Engineering, King Abdullah University of Science and Technology (KAUST), Jeddah, Saudi Arabia

*Correspondence to*: Jason P. Evans (jason.evans@unsw.edu.au)

**Abstract.** In this study we have examined the ability of a regional climate model (RCM) to simulate the extended drought that occurred throughout the period 2002 through 2007 in southeast Australia. In particular, the ability to reproduce the two drought peaks in 2002 and 2006 was investigated. Overall the RCM was found to reproduce both the temporal and the spatial structure of the drought related precipitation anomalies quite well, despite using climatological seasonal surface characteristics such as vegetation fraction and albedo. This result concurs with previous studies that found that about two thirds of the precipitation decline can be attributed to ENSO. Simulation experiments that allowed the vegetation fraction and albedo to vary as observed illustrated that the intensity of the drought was underestimated by about 10% when using climatological surface characteristics. These results suggest that in terms of drought development, capturing the feedbacks related to vegetation and albedo changes may be as important as capturing the soil moisture-precipitation feedback. In order to improve our modelling of multi-year droughts the challenge is to capture all these related surface changes simultaneously, and provide a comprehensive description of land surface-precipitation feedback during the droughts development.

## 1 Introduction

Feedbacks in the climate system have the potential to exacerbate or alleviate extremes such as droughts. Over the land surface, feedbacks to precipitation are often mediated through changes in the soil moisture. These feedbacks can involve a number of processes and can be measured in a variety of ways (see Seneviratne et al., 2010). The multiple mechanistic pathways and the non-linear nature of the connection between the smoothly varying soil-moisture field and highly episodic precipitation makes the feedback strength difficult to quantify with confidence. While some studies have used observations to quantify the soil moisture – precipitation feedback (Findell and Eltahir, 1999; Taylor et al., 2011, 2012; Catalano et al., 2016), more common is the use of model experiments to isolate and allow quantification of this behaviour (Schar et al., 1999; Koster et al., 2006; Seneviratne et al., 2013; Hirsch et al., 2014). Other slowly varying surface variables that have been found to provide feedbacks

to precipitation include albedo (Charney et al., 1975; Lofgren, 1995; Zaitchik et al., 2007; Teuling and Seneviratne, 2008; Meng et al., 2014b) and vegetation (Pielke et al., 1998; Zeng and Neelin, 2000; Wang et al., 2006; Meng et al., 2014a). These feedbacks act on different time scales and can subdue or reinforce the feedback from the soil moisture field. This emphasises the difficulty in identifying feedback mechanisms when changes to all these surface fields are occurring simultaneously.

The influence of land – atmosphere feedbacks may be particularly important in the development of extreme events such as droughts. Such a connection has been recognised since at least Charney et al. (1975) who, using a Global Climate Model (GCM), found that a change in surface albedo caused by a decrease in vegetation would cause a decrease in rainfall over the Sahara. This provided a positive feedback that enhanced drought conditions. Charney et al. (1977) extended this work, finding

evidence for similar positive feedbacks in other semi-arid regions. Since these pioneering investigations, climate models have continued to improve and many studies into the sensitivities of climate models to land surface conditions have been performed.

A number of these studies have focused on how the surface feedbacks effect the development of particular locations or drought events. For instance, Oglesby and Erickson (1989) used a GCM to examine the influence of soil moisture on drought in north

America, also finding a positive feedback that enhances drought conditions. Hong and Kalnay (2000) used an RCM to investigate the role of local feedbacks in the development of the Texas USA drought in 1998. They found that the surface feedbacks were responsible for up to 30% of the precipitation deficit during the drought. Schubert et al. (2004) investigated causes of the North American dustbowl drought in the 1930s. They attributed 50% of the precipitation deficit to soil moisture-precipitation feedbacks. Zaitchik et al. (2007) examined the surface influence on a drought that occurred in the Middle East in

1999. Using a Regional Climate Model (RCM) they found that vegetation and albedo changes had clear effects on the surface fluxes and Planetary Boundary Layer (PBL) growth, but limited impact on the precipitation decrease (up to 4%) compared to a normal year. Wu and Zhang (2013) performed a RCM investigation of soil moisture feedback on the 1999 drought in northern China, finding that the feedback accounted for up to 50% of the precipitation decline in some places. Zaitchik et al. (2012) investigated the surface feedback on the southern Great Plains USA drought of 2006. They found that the precipitation decline

during drought development increased by ~10% due to the feedback. Finally, Meng et al. (2014a, 2014b) examined the role of changes in surface albedo and surface vegetation on the development of the 2002 drought in south east Australia. They found that the precipitation reduction was enhanced by up to 20% and 10% due to surface albedo and vegetation changes respectively. Importantly, they identified differences in time scales over which changes in vegetation occur compared to changes in soil moisture or albedo, with the relatively slow vegetation changes tending to dampen the positive soil moisture –

precipitation feedback. All of these studies showed that the land surface – precipitation feedbacks play an important role in drought development. However, the strength of this role is both space and time dependent.

In general terms, mechanisms that produce soil moisture-precipitation feedback involve a change in energy partitioning at the surface that subsequently changes the evolution of the Planetary Boundary Layer (PBL) and the likelihood of triggering

precipitation. That is, a decrease in soil moisture leads to more energy being used for sensible heating, an increase in the PBL height with a related decrease in the moist static energy density, resulting in a decreased likelihood of triggering precipitation and a further reduction in soil moisture. Changes in other surface characteristics, such as albedo and vegetation cover, can also change the surface energy partitioning, and produce a similar chain of affects that result in a feedback on the soil moisture

conditions. Unlike soil moisture-precipitation feedbacks however, the relationship between changes in these other surface characteristics and the surface energy partitioning can be quite complex. For example, an increases in albedo will reduce the net radiation at the surface but how will this reduction in available energy be partitioned between the surface fluxes? Similarly, a reduction in vegetation cover will mean more exposed soil from which water can evaporate quickly following a rain storm, but a reduction in the vegetated area that can continue transpiring through a dry spell (up to a point). So vegetation changes

have a time varying impact on surface energy partitioning but what is the cumulative impact on the surface energy fluxes? In reality soil moisture, albedo and vegetation all change simultaneously. Here we explore the impact of these changes on the development of drought in south-east Australia.

 

The Murray-Darling Basin (MDB) is Australia's largest river system (Figure 1). It covers a catchment area of ~1,000,000 km$^2$

or ~14% of Australia. Unlike most other major basins of the world, the MDB is predominantly semi-arid with a very low ratio of discharge to precipitation and exhibits high interannual hydrologic variability. The MDB is Australia's "food basket" (Nicholls, 2004), accounting for 40% of Australia's agricultural production. The 1,500,000 hectares under irrigation for crops and pastures represents 70% of the total area under irrigation in Australia, with more than 80% of the divertible surface water resource being consumed locally. From 2001 to 2008 the worst drought in recorded history was experienced in the MDB, with

the 7-year averaged rainfall being the lowest since 1900 (Potter and Chiew, 2011). An investigation into the causes and impacts of the drought, referred to as the "Millennium drought", was performed by van Dijk et al. (2013). They found that large scale climate modes, particularly in the Pacific ocean, could explain about two thirds of the precipitation deficit, leaving about 30% of the deficit unexplained and potentially related to local feedbacks (Evans et al., 2011). This study extends the work of Meng et al. (2014a, 2014b) who examined the effects of albedo changes and vegetation changes in isolation from one another. In

reality factors such as soil moisture, albedo and vegetation, change simultaneously but over different time scales during drought development and intensification. This study adds value to previous work by examining the combined evolution of these factors and the resulting feedback on precipitation during the evolution of the Millennium drought.

 

## 2 Data Description

### 2.1 Precipitation and Temperature data

Gridded precipitation and near surface air temperature products were used to evaluate the RCM. These 5 km resolution gridded products are interpolated from station measurements as part of the Australian Water Availability Project (AWAP) (Jones et al., 2009). These data have been widely used in a number of hydrological and climate studies in south-east Australia (e.g. Cai

et al., 2009; Olson et al., 2016; Teng et al., 2015). Here the 5 km resolution products were interpolated to 10 km resolution to enable direct comparison with the RCM results. Figure 2 shows the 12-month smoothed precipitation anomaly in the Murray and Darling River basins. Strong minima can be seen in 2002 and 2006 in both basins. The millennium drought spans this entire period, with 2002 and 2006 being separate peaks in meteorological drought conditions.

## 2.2 Albedo data

The default albedo product used in the RCM was derived from the Advanced Very High Resolution Radiometer (AVHRR) based upon monthly mean clear-sky, snow free surface broadband albedo data retrieved between 1985 and 1991 (Csiszar, 2009). It is applied as a monthly climatology. The observed albedo data used was produced using nadir BRDF (bidirectional reflectance distribution function) adjusted reflectances at 1,000 m spatial resolution and 8-day intervals with 16-Day data composites (MCD43B4.005). Data was downloaded from the MODIS Land Mosaics for Australia in the Water Resources Observation Network (WRON) of Australia's Commonwealth Scientific and Industrial Research Organisation (CSIRO). The original data used to produce the MODIS Land Products for Australia were supplied by the Land Processes Distributed Active Archive Center (LPDAAC), located at the U.S. Geological Survey (USGS) Earth Resources Observation and Science Center (EROS). Further details on the data are provided in Paget and King (2008) with additional quality control described in Meng et al. (2014b). Albedo anomalies are shown in Figure 3. The default albedo has the same seasonal anomalies every year (with slight differences due to seasonal snow cover), while the observed albedo starts lower and moves to higher values as the drought conditions worsen.

## 2.3 Vegetation Fraction data

The default vegetation fraction dataset used in the RCM was derived from Advanced Very High Resolution Radiometer (AVHRR)-based monthly mean normalized difference vegetation index (NDVI) data between 1985 and 1991. Details of the data are given in (Gutman and Ignatov, 1998). Like albedo, it is applied as a monthly climatology. The observed vegetation fraction data used in the simulation experiments was produced using the nadir BRDF adjusted data from the combined Terra–Aqua MODIS product (MCD43A4.005). A linear unmixing methodology was used in the derivation of vegetation fraction (Guerschman et al., 2009). Further details on the processing and analysis of this observation record are provided in Meng et al. (2014a). Vegetation fraction anomalies are shown in Figure 3. As can be seen, the default dataset is dominated by the seasonal cycle, while the observed vegetation fraction is dominated by inter-annual changes, with lower values associated with more extreme drought conditions.

## 3 Regional Climate Model Simulations

The RCM used here was built within the Weather Research and Forecasting (WRF) model framework. The WRF model is a widely used atmospheric model maintained at the National Center for Atmospheric Research (NCAR) in the United States.

The Advanced Research WRF is a nonhydrostatic, terrain-following, dry hydrostatic-pressure coordinate model designed to simulate or predict regional-scale atmospheric circulation. WRF has been comprehensively evaluated across numerous investigations over southeast Australia and has been found to perform well (Cortés-Hernández et al., 2015; Evans and McCabe, 2010; Evans and Westra, 2012).

Version 3.1.1 (Skamarock et al., 2008) was applied in this study using the following physics schemes: the Kain–Fritsch cumulus physics scheme, the WRF single moment 5-class microphysics scheme, the Dudhia shortwave radiation scheme, the Rapid Radiative Transfer Model (RRTM) longwave radiation scheme, the Yonsei University boundary layer scheme, Monin–Obukhov surface layer similarity, and the Noah land surface scheme. The model simulation uses 6-hourly boundary conditions from the National Centers for Environmental Prediction (NCEP)–NCAR reanalysis project (NNRP - Kalnay et al., 1996) with an outer 50-km resolution nest and an inner 10-km resolution nest that covers southeast Australia (Fig. 1). Both nests used 30 vertical levels (see Evans and McCabe (2010) for further details of the model setup).

The RCM simulations in this study began at the start of the year 2000 using the climate state produced by running the model for 15 years (1985–99) to spin up the soil moisture states in a coupled environment. Four simulations were performed: one using the WRF default albedo and default vegetation fraction (WRF_CTL); one using the default vegetation fraction and observed albedo (WRF_ALB); one using the default albedo and observed vegetation fraction (WRF_VEG); and one using observed albedo and observed vegetation fraction (WRF_BOTH). The Noah land surface scheme is described in (Chen and Dudhia, 2001). In this implementation the green vegetation fraction is used to determine the fraction of a grid cell that is covered by vegetation vs bare soil. It has a direct impact on the partitioning of evaporation between soil evaporation, canopy evaporation and transpiration. The albedo changes the amount of upward shortwave radiation and hence the energy available for use in other surface energy fluxes.

The Murray and Darling River basins are the focus regions of this study (Figure 1). The Great Dividing Range to the east of both basins is a temperate zone that captures most of the precipitation that supplies the rivers. The Darling basin has a subtropical region in the north but generally transitions through semi-arid grasslands towards desert in the west. The Murray basin is dominated by the temperate region in the east and south, and contains grasslands in the north-west. In terms of rainfall, the Murray basin is more consistently wet with winter dominant precipitation, while the Darling basin has large dry areas with summer dominant precipitation.

# 4 Results

## 4.1 Evaluation of simulations

The RCM simulations were first evaluated against the AWAP observations to ensure a reasonable representation of the regions climate is obtained. A summary of the evaluation results for each river basin is given in Table 1. Note that two factors are being tested here. First, the default albedo and vegetation fraction datasets represent climatological conditions in the late 1980s not at the time of interest. Substantial changes in the land surface may have occurred over the intervening 20 years. There may also be some offset between the AVHRR (default) and MODIS (observed) sensors. Most of the bias between the default and observed datasets may be due to this temporal and sensor mismatch. Second, the default datasets do not capture the inter-annual variability associated with drought development. This mismatch between the default and observed datasets will have some impact on the RMSE and pattern correlation statistics. The effect of this inter-annual variability is the focus of sections 4.2 and 4.3 which examine changes in time within each simulation, thus the influence of between simulation biases are largely removed.

In terms of 2 m air temperature, the simulations improve in all respects as more of the observed surface conditions are included, such that WRF_BOTH produces the best statistics. The results for precipitation are more mixed, with the inclusion of observed vegetation changes (WRF_VEG) producing the lowest bias. The addition of observed albedo in WRF_BOTH leads to a deterioration of the bias averaged over the river basins. It is worth noting that the simulations show very little difference in either the pattern or anomaly correlations, indicating that the changes in albedo and vegetation have little effect on the average precipitation or temperature spatial distribution, which is strongly influenced by topography.

The spatial distribution of the bias is shown in Figure 4. The mean annual precipitation from the AWAP observations is shown along with the precipitation biases for each of the simulations. The saturation/intensity of the colours in the bias plots show the bias as a percentage of the annual precipitation, while the hue presents the bias as a total in mm/month. Grey areas indicate that the bias is less than 10% of the annual precipitation. For all simulations the biases are generally less than 20% throughout both river basins. It can be seen that the WRF_BOTH simulation covers more of the southern (Murray) river basin with biases of less than 10%, indicting a better match over more of the region than the other simulations.

**Table 1: Summary evaluation statistics for monthly temperature and precipitation fields simulated by each experiment compared to AWAP observations.**

| | | Temperature (K) | | | | Precipitation (mm) | | | |
|---|---|---|---|---|---|---|---|---|---|
| | | WRF_CTL | WRF_ALB | WRF_VEG | WRF_BOTH | WRF_CTL | WRF_ALB | WRF_VEG | WRF_BOTH |
| Murray | Bias | 0.90 | 0.65 | 0.51 | 0.27 | -4.87 | -5.91 | -4.26 | -5.40 |
| | RMSE | 1.42 | 1.26 | 1.21 | 1.12 | 21.4 | 21.1 | 21.4 | 21.1 |
| | Pattern Correlation | 0.99 | 0.99 | 0.99 | 0.99 | 0.87 | 0.86 | 0.87 | 0.87 |
| | Anomaly Correlation | 0.94 | 0.94 | 0.94 | 0.94 | 0.66 | 0.66 | 0.66 | 0.66 |
| Darling | Bias | 1.24 | 0.80 | 0.82 | 0.38 | -3.10 | -6.19 | -2.23 | -5.24 |
| | RMSE | 1.56 | 1.24 | 1.33 | 1.07 | 27.9 | 28.0 | 27.8 | 27.8 |
| | Pattern Correlation | 0.99 | 0.99 | 0.99 | 0.99 | 0.82 | 0.82 | 0.83 | 0.82 |
| | Anomaly Correlation | 0.95 | 0.95 | 0.95 | 0.95 | 0.55 | 0.55 | 0.56 | 0.56 |

## 4.2 Representation of drought

The time series of 12-month running average precipitation for AWAP observations and simulation experiments, averaged over each of the river basins, are presented in Figure 5. In agreement with the biases shown previously we see that the simulations tend to underestimate the amount of precipitation in both basins. Importantly, the simulations reproduce the two main precipitation minima in 2002 and 2006/2007 very well, with similar rainfall declines to the observations indicating that the simulations are able to capture the drought dynamics quite well. It can be seen that the differences between the simulations are small compared to the precipitation declines leading to the drought minima. Figure 6 provides a clearer perspective of the difference between the experimental simulations and the control simulation. Here we see that the albedo increases tend to produce less precipitation than the control run, while the vegetation changes tend to produce more precipitation. The combined experiment (WRF_BOTH) result resembles a non-linear combination of the two individual change experiments. We also note that the largest differences between WRF_BOTH (or WRF_ALB) and WRF_CTL occur in 2007 (Figure 6), indicating that the observed albedo increases tend to delay the drought recovery.

The spatial distribution of precipitation change for the droughts in 2002 and 2006 are shown in Figure 7 and Figure 8 respectively. For the 2002 drought the simulations are able to capture the extent and magnitude of this precipitation decline across the majority of both river basins. The simulations do not do as good a job of reproducing the spatial pattern of declines during the 2006 drought (Figure 8). In the Murray basin, fairly large declines were observed throughout most of the basin,

with a maximum in the southeast. The simulations also produced maximum declines in the southeast but produced weaker declines throughout most of the rest of the basin. The Darling basin is observed to have precipitation declines across the south of the basin and in a band extending up to the north/northwest, with small precipitation increases either side. The simulations produce precipitation declines in the southern part of the basin but struggle to produce declines as large as the observed in the band to the north/northwest. While it can be difficult to distinguish between the simulations, here we see the BOTH simulation getting closest to the north/northwest declines.

While not surprising, it is worth noting that in the observations (and hence in our experiments), the drought related vegetation and albedo changes are highly anti-correlated. Figure 9 shows the bivariate joint probability distribution of the albedo and vegetation changes for each of the droughts. Generally a decrease in vegetation is associated with an increase in albedo. It can be seen that larger overall changes in vegetation and albedo are associated with the 2002 drought compared to the 2006 drought. While in both cases the linear regression relationship is significant at the 0.99 level, the 2002 drought has much larger albedo changes for each unit of vegetation fraction change. This reflects the additional role of soil moisture changes affecting albedo. In the 2002 drought the soils dried substantially from relatively wet in 2000 to dry in 2002 (Liu et al., 2009). In the 2006 drought however, the soils transitioned from relatively dry to even drier, with a much smaller impact on albedo.

The relationship between albedo and vegetation changes with changes in the precipitation is explored in Figure 10 and Figure 11. The bivariate joint probability between the albedo change and the precipitation change for each drought is shown in Figure 10. Here we can see that the majority of albedo increases are associated with precipitation decreases. When the vegetation changes are also included (WRF_BOTH), we tend to see that all levels of albedo changes are associated with larger precipitation decreases. The effects of the vegetation changes varies much more between droughts, as shown in Figure 11. In the 2002 drought we see that almost everywhere there are decreases in vegetation fraction that are associated with decreases in precipitation. When albedo changes are also included (WRF_BOTH) we see a somewhat random redistribution of precipitation changes. In the 2006 drought, vegetation changes are more centred on no change. When albedo changes are also included (WRF_BOTH) there is a clear decrease in the frequency of precipitation increases and an increase in large precipitation decreases.

### 4.3 Feedback mechanisms

Here we examine the surface energy budget and potential feedback mechanisms during the development of both droughts. In the Murray basin drought years have less latent heat and more sensible heat as expected (Figure 12a and b). The effect of allowing albedo and vegetation fraction to vary as observed is shown more clearly in Figure 12c and d which shows the difference in these changes during drought development in each experiment compared to the CTL. For the 2002 drought (Figure 12c) when only the observed albedo increase is included there is a decrease in surface net radiation (Rnet) compared

to CTL. This decrease is used entirely to decrease the sensible heating (SH). This is typical of a water limited environment where water availability controls the latent heat (LH) not the energy. In the 2006 drought the decrease in Rnet is split between LH and SH. Like the 2002 case, these changes are controlled by the water availability. The higher albedo produces a negative feedback on precipitation that is persistent over many years (Figure 6) and results in lower root zone soil moisture, hence less water available for evapotranspiration, compared to CTL.

In the 2002 drought when only the observed vegetation fraction decrease is included, there is a large decrease in latent heating and a somewhat compensating increase in sensible heating. In this case the vegetation fraction starts higher than the CTL and more transpiration of water from the root zone occurs. During the drought year the vegetation fraction has reduced and the soil moisture depleted such that similar LH occurs in both the VEG and CTL cases. Hence the decrease during drought development is greater in the VEG case. In the 2006 drought a similar but damped response occurs since the available soil moisture is lower in 2005 than 2000.

When both the observed albedo increases and vegetation fraction decreases are included, the surface energy balance response is a non-linear combination of the two previous cases. It is worth noting that while the Rnet decrease is similar during the development of the 2002 and 2006 droughts, the change in energy partitioning compared to CTL, is three times larger in 2002 than 2006. The lower available soil moisture before the 2006 drought onset has damped the surface flux changes.

These damped fluxes lead to different feedbacks operating during the development of each drought. Using the methodology in Meng et al. (2014b) to examine the relative contribution to decreases in the moist static energy density of the PBL we see that the 2006 drought (Figure 13) has a smaller total area with the feedback operating and the dominant cause is a decrease in the total turbulent heat flux. This differs from the 2002 drought (figure 14 in Meng et al., 2014b) where the dominant cause is an increase in the PBL height. These feedbacks act relatively quickly and are mostly related to changes in albedo and the current soil moisture state.

To account for the different time scales associated with albedo and vegetation changes the methodology in Meng et al. (2014a) can be used to identify the presence of fast physical feedbacks associated with albedo and soil moisture changes as discussed above, and slower vegetation related changes that impact the strength of the fast feedbacks. The Figure 14 shows when the fast physical and slow biological mechanisms are active during the 2006 drought and is comparable to figure 12 in Meng et al. (2014a) which shows the same thing for the 2002 drought. The findings confirm that the damped surface fluxes during the development of the 2006 drought results in less area exhibiting the feedbacks compared to 2002, particularly the slow biological feedback. It also concurs with the finding in Meng et al. (2014a) that the fast feedback is less likely to occur if the slow feedback is present (relatively few orange areas) as it acts to reduce the soil moisture changes.

## 5 Discussion

The response of albedo and vegetation to the development of each of the droughts differs. The 2002 drought occurs after two years of declining precipitation from a normal/wet year in 2000, while the 2006 drought occurs after one year of declining precipitation that follows a normal year at the end of a dry period. In the Darling basin the 2002 drought is accompanied by a steadily increasing albedo and a decrease in vegetation fraction that is delayed by a year (Figure 3). Neither the albedo nor the vegetation fraction recovers to pre-2002 drought levels before the 2006 drought arrives. The 2006 drought is accompanied by a smaller increase in albedo and a small decline in vegetation compared to the 2002 drought. This delayed vegetation response was explored in Meng et al. (2014a) who showed that while albedo responses and feedbacks occur on relatively short time scales (~1 month), large scale changes in vegetation occur over longer time scales (~1 year) and hence become more important in multi-year droughts.

In the Murray basin, the 2002 drought is accompanied by a steady albedo increase that occurs in summer only, while the related vegetation fraction decrease continues through into 2003. The 2006 drought sees a similar decrease in albedo but the vegetation fraction experiences a one year decrease with a similar magnitude to that experienced in 2002. It should be noted that in the Murray basin during the peak drought years, an entire phenological cycle is skipped and replaced by a steady decline in vegetation. This ability for vegetation to skip phenological cycles during drought years is an adaptation found commonly in Australian arid or semi-arid zones (Broich et al., 2014). Here we see that even temperature zone species are able to do this to an extent in extreme drought years.

The spread of the experiments is much smaller than the drought related precipitation declines (Figure 5) indicating that the droughts are mostly driven by external processes, with albedo and vegetation changes producing a smaller effect on precipitation. This concurs with van Dijk et al. (2013) who attribute about two thirds of the rainfall deficit to ENSO. Comparing Figure 5 and Figure 6 allows an estimate of the role played by changing albedo and vegetation on the total precipitation decline in each basin. In the Murray basin, from 2000 until 2002, the precipitation declined by ~18 mm/month and the WRF_BOTH simulation declined an extra 2 mm/month compared to the WRF_CTL simulation. From 2005 to 2006 the precipitation declined ~10 mm/month and the WRF_BOTH simulation declined an extra 1 mm/month compared to the WRF_CTL simulation. In the Darling basin, from 2000 until 2002, the precipitation declined by ~20 mm/month and the WRF_BOTH simulation declined by an extra 2 mm/month compared to the WRF_CTL simulation. From 2005 to 2006 the precipitation declined 8 mm/month and the WRF_BOTH simulation declined by an extra 1 mm/month compared to the WRF_CTL simulation. In each case the albedo and vegetation changes combined can account for ~10% of the precipitation decline. This is double the contribution found for albedo and vegetation changes to a drought in the Middle East (Zaitchik et al., 2007).

In all of the simulations in this study the soil moisture-precipitation feedback is present. Previous studies that explicitly quantified the magnitude of the soil moisture-precipitation feedback on the development of drought found that the feedback accounted for anything from 10% to 50% of the precipitation decline (Hong and Kalnay, 2000; Schubert et al., 2004; Wu and Zhang, 2013; Zaitchik et al., 2012). Here we find that the addition of vegetation fraction and albedo changes add a further 10% to the drought related precipitation decline. While the wide range found for the soil moisture-precipitation feedback appears to be location and model dependent, the fact that the impact of albedo and vegetation changes falls within this range suggests that they should not be ignored. In reality, soil moisture, albedo and vegetation changes all occur simultaneously, albeit over different time scales. The challenge for the land surface modelling community is to predict changes in all these surface characteristics in a physically consistent way, over drought relevant time scales, that will allow simulation of the total surface feedbacks and hence produce more realistic drought development within climate models.

## 6 Conclusions

In this study we have examined the ability of a regional climate model (WRF) to simulate the extended drought that occurred from 2002 through 2007 in southeast Australia. In particular, the ability to reproduce the two drought peaks in 2002 and 2006 was investigated. Overall the RCM was found to reproduce both the temporal and the spatial structure of the drought related precipitation anomalies quite well, despite using climatological seasonal surface characteristics such as vegetation fraction and albedo. This concurs with previous studies that found that about two thirds of the precipitation decline can be attributed to ENSO. Satellite based observations show substantial inter-annual variability in albedo and vegetation fraction, particularly in drought periods. These variations were found to be highly anti-correlated, showing that in reality simultaneous changes in both quantities are occurring and need to be accounted for. Experiments that allow for the vegetation fraction and albedo to vary as observed shows that the intensity of the drought is underestimated by about 10% when using climatological surface characteristics. These results suggest that in terms of drought development, capturing the feedbacks related to vegetation and albedo changes may be as important as capturing the soil moisture-precipitation feedback. In order to improve our modelling of multi-year droughts, the challenge is to capture all these related surface changes simultaneously, and provide a comprehensive land surface-precipitation feedback during the droughts development.

## 7 Acknowledgments

This work was funded by the Australian Research Council as part of the Discovery Project DP0772665 and Future Fellowship FT110100576. This work was supported by an award under the Merit Allocation Scheme on the NCI National Facility at the ANU. This paper is submitted to the special issue "Observations and Modeling of Land Surface Water and Energy Exchanges Across Multiple Scales" in honor of Professor Eric F. Wood. Eric has inspired a generation of researchers in hydrology and related sciences, including myself.

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

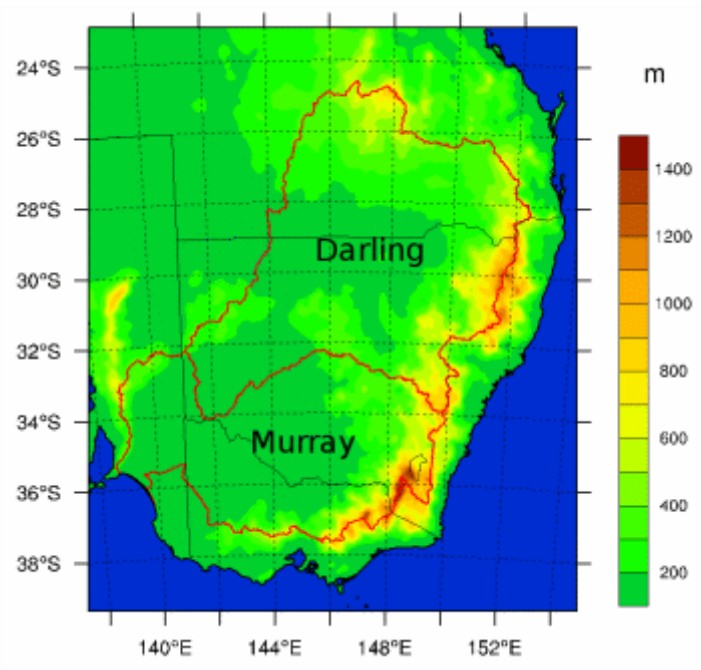

5    **Figure 1: Model domain showing the Murray and Darling River basins.**

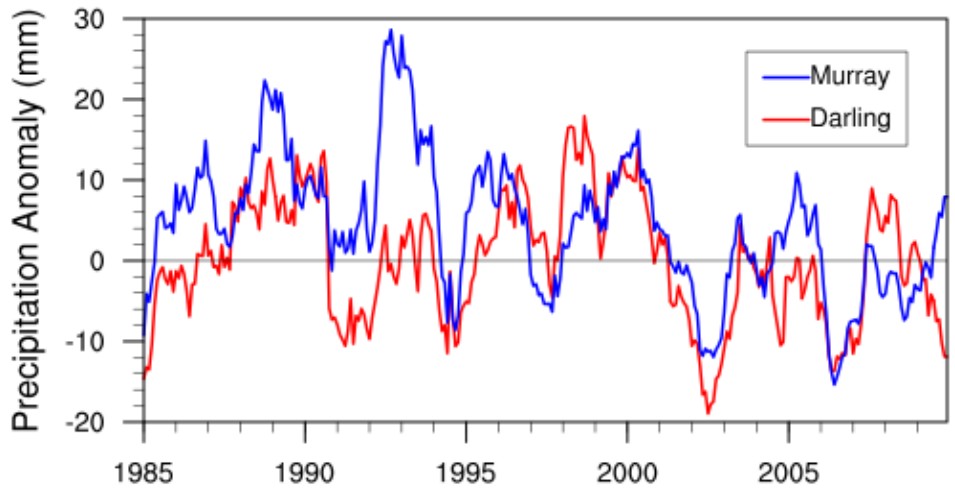

**Figure 2: 12-month smoothed precipitation anomaly for the Murray and Darling River basins.**

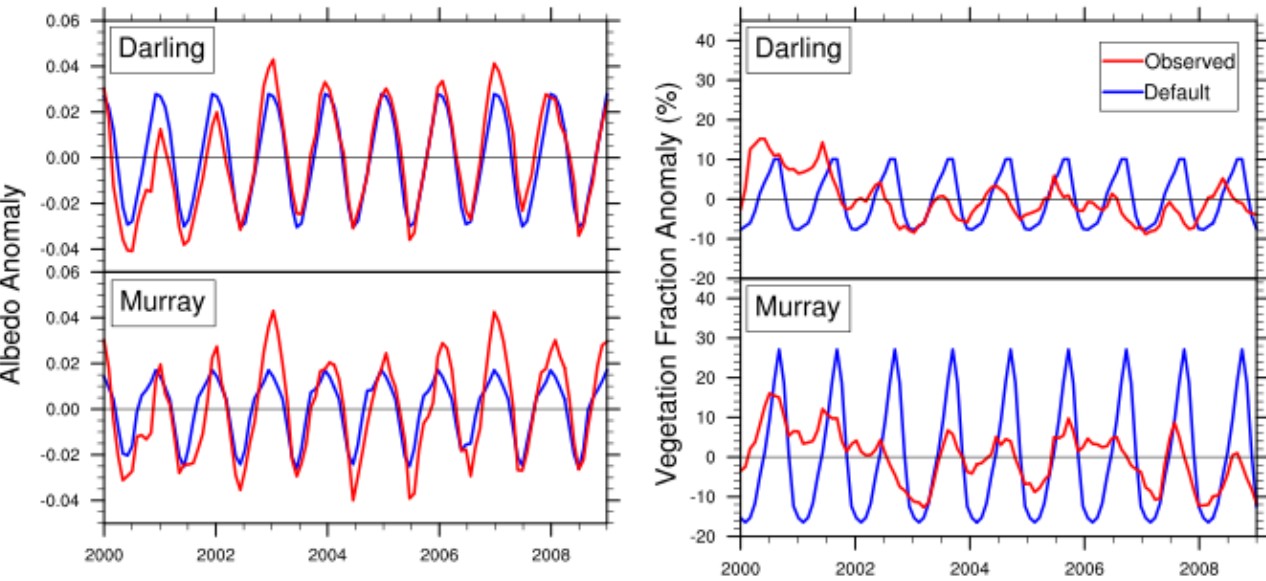

**Figure 3: Albedo and vegetation fraction anomalies.**

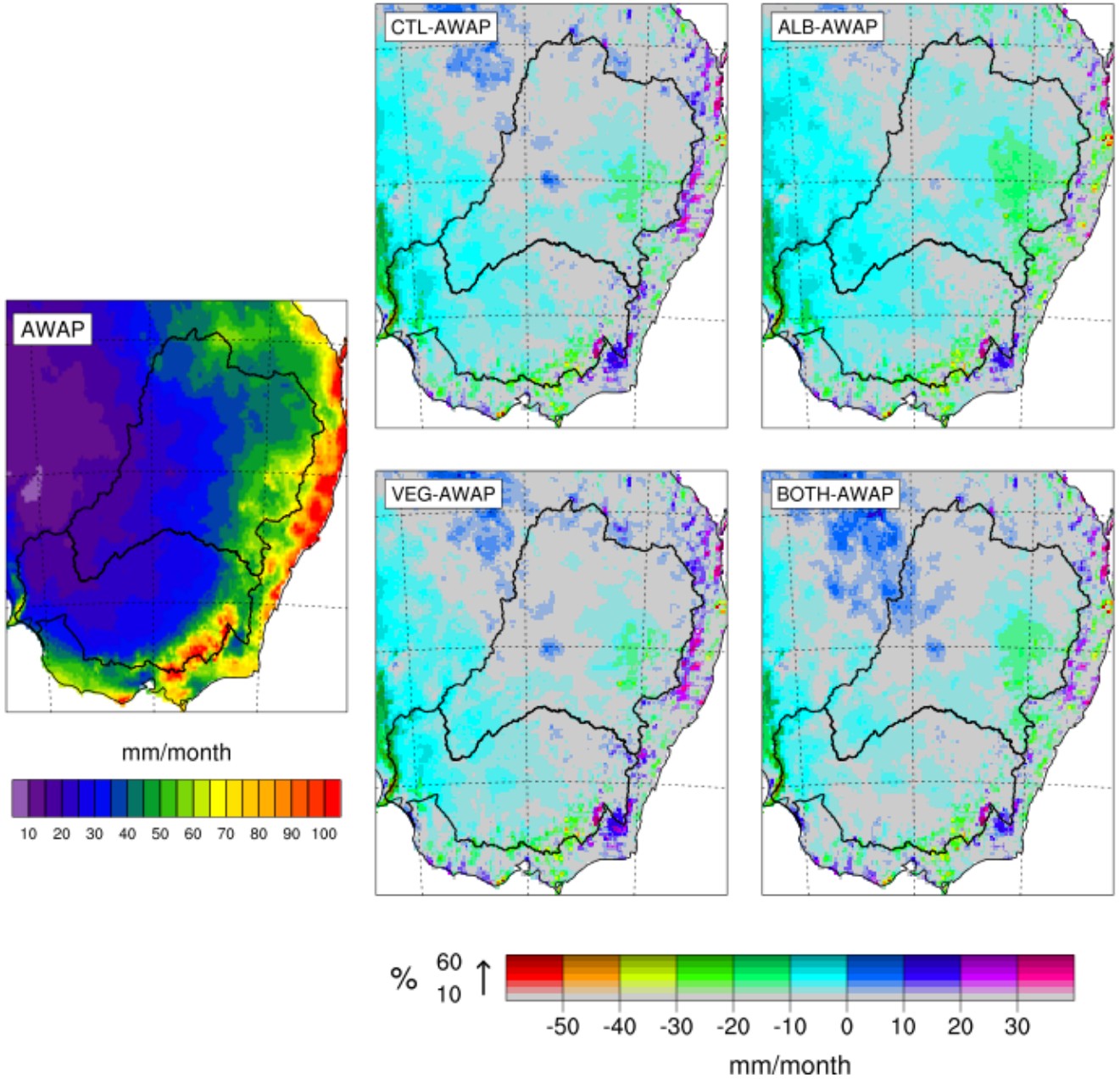

**Figure 4: Annual precipitation and precipitation bias of each model simulation. The hue of the colours gives the bias in mm/month while the saturation/intensity gives the bias in percentage terms. Grey areas have less than 10% bias.**

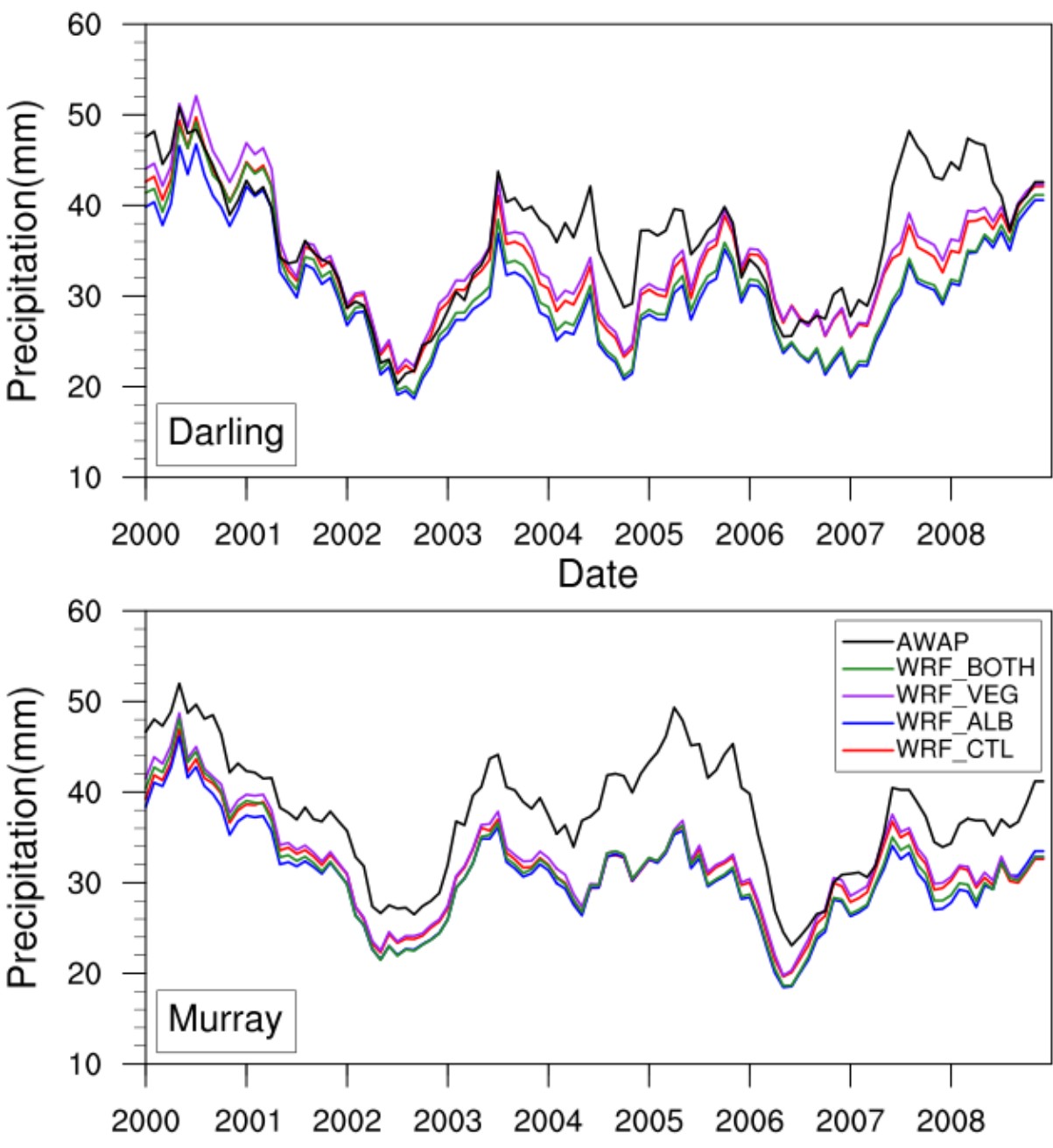

**Figure 5: 12-month running average precipitation (mm/month) for each model simulation and observations, averaged over each river basin.**

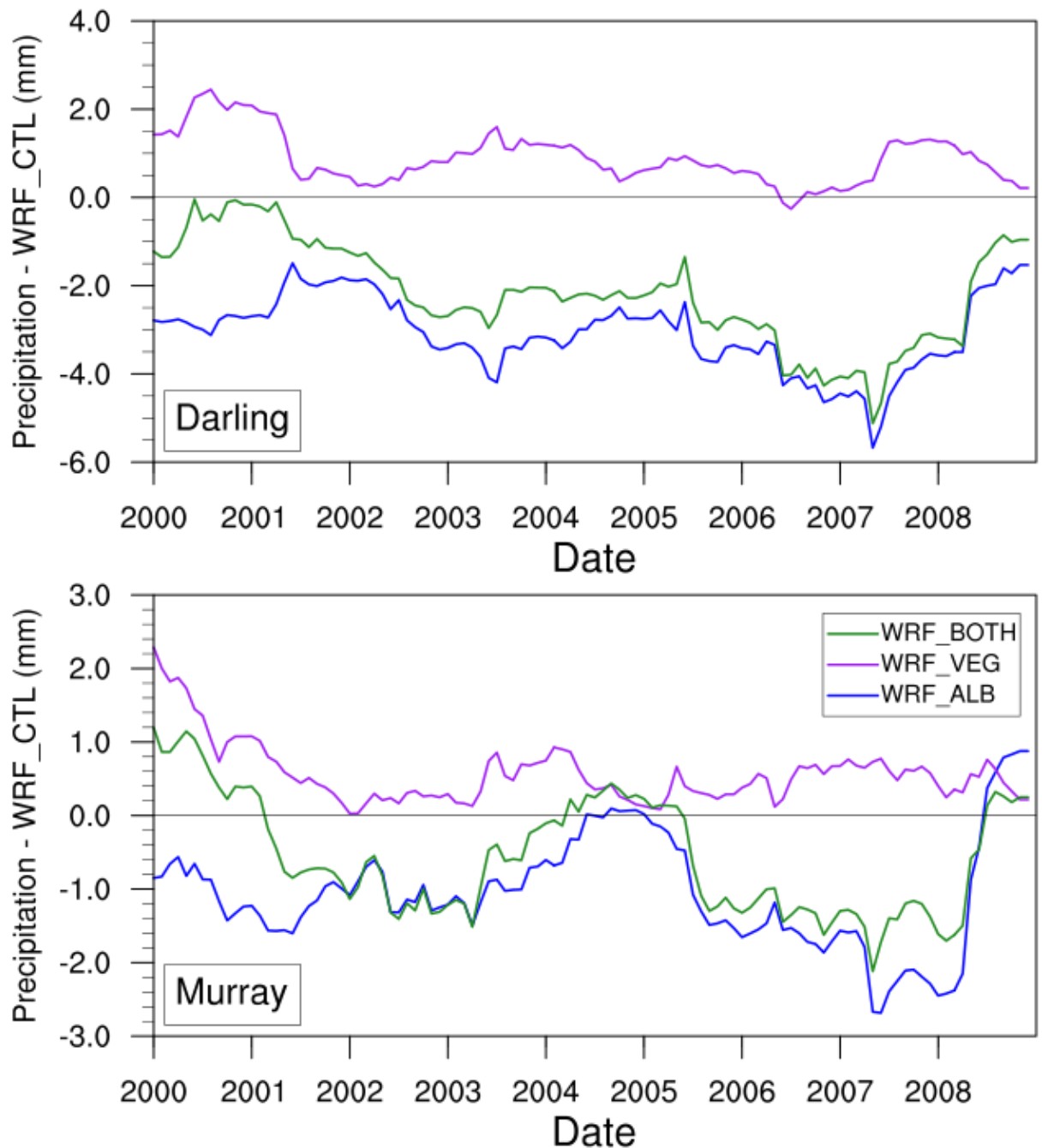

**Figure 6: The difference in 12-month running average precipitation (mm/month) between each model simulation and WRF_CTL, averaged over each river basin.**

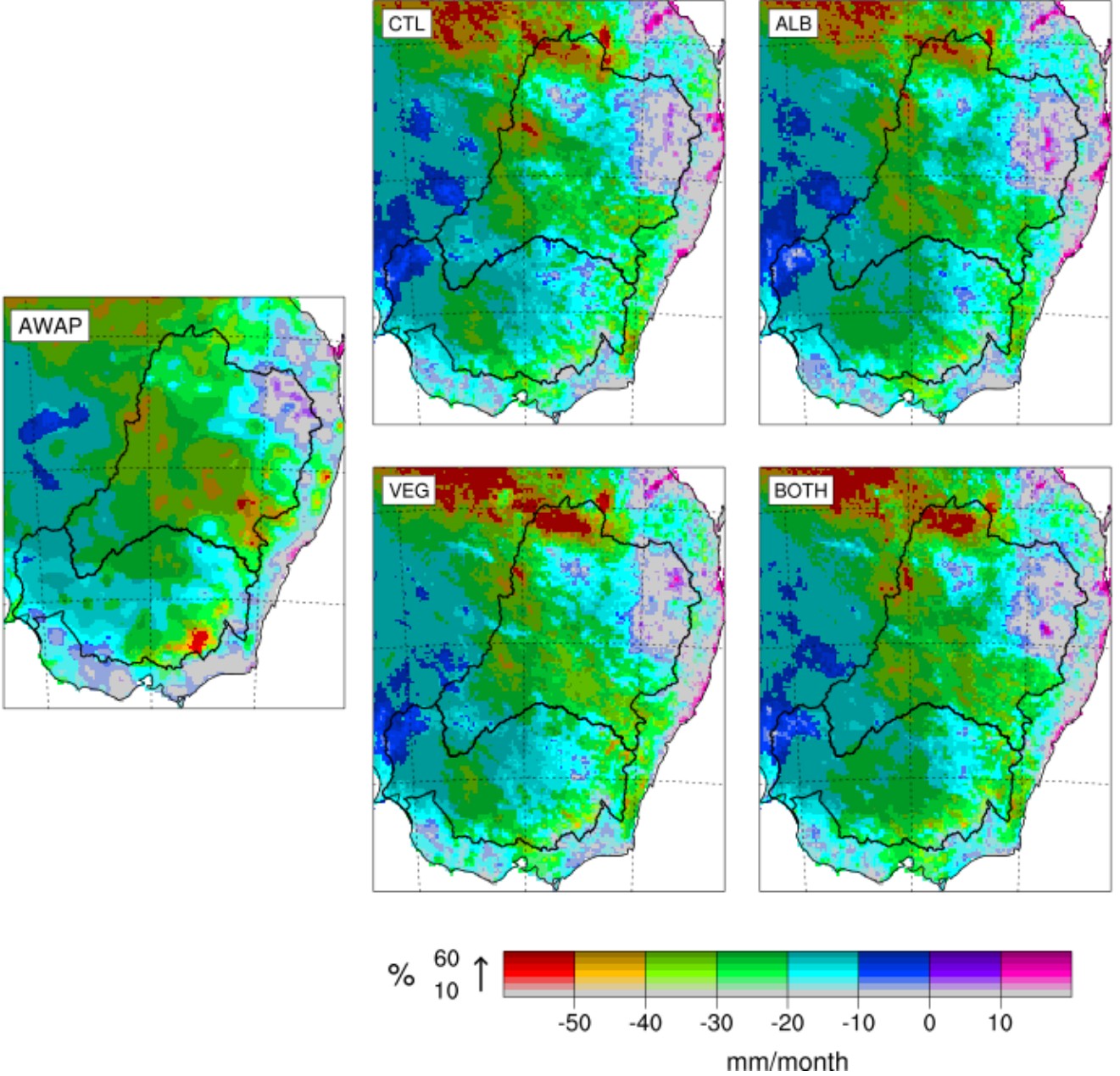

**Figure 7: The precipitation change from 2000 (normal/wet) to 2002 (drought) for each model simulation and observations. The hue of the colours gives the change in mm/month while the saturation/intensity gives the change in percentage terms. Grey areas have less than 10% change.**

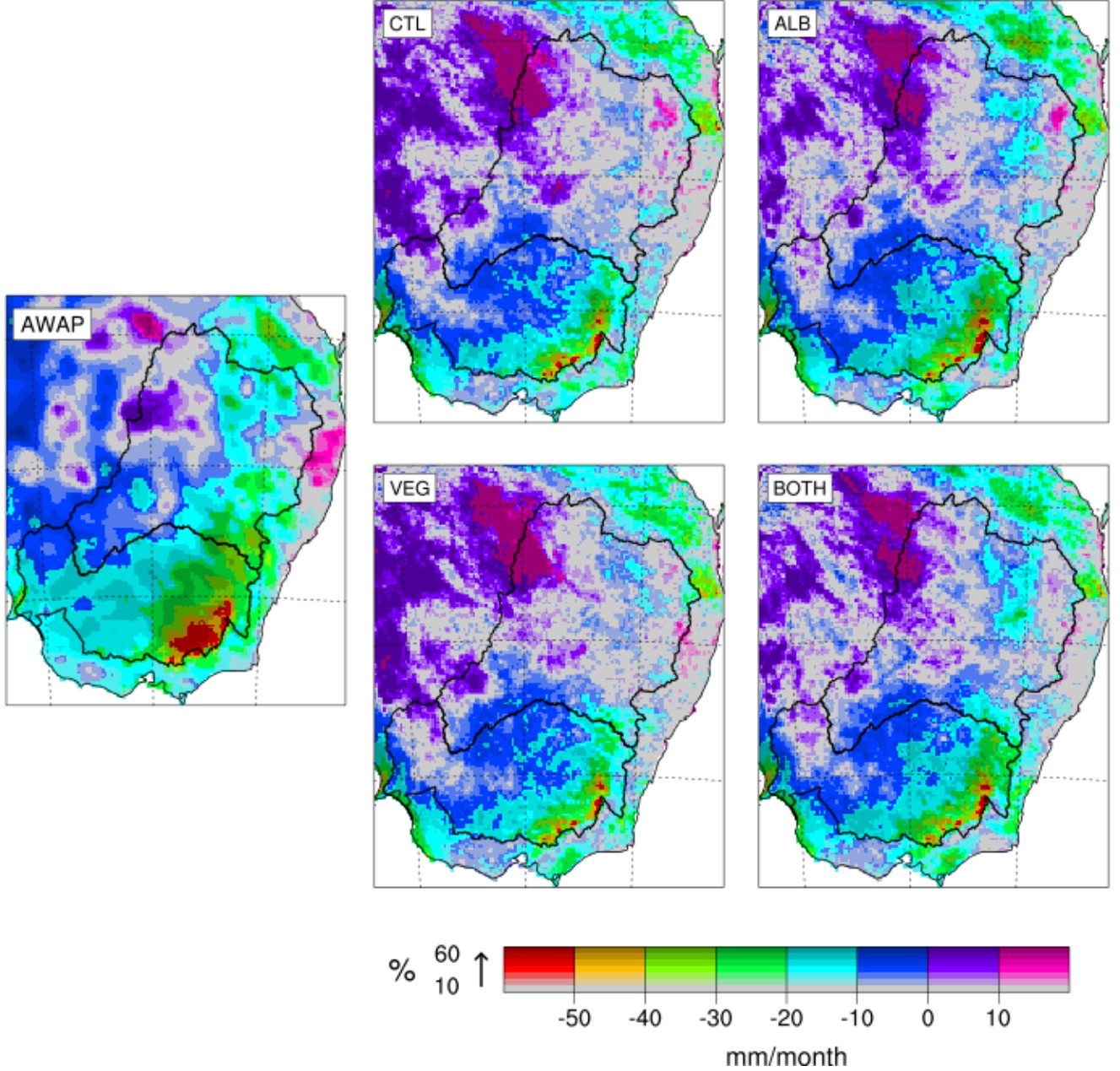

**Figure 8: The precipitation change from 2005 (normal) to 2006 (drought) for each model simulation and observations. The hue of the colours gives the change in mm/month while the saturation/intensity gives the change in percentage terms. Grey areas have less than 10% change.**

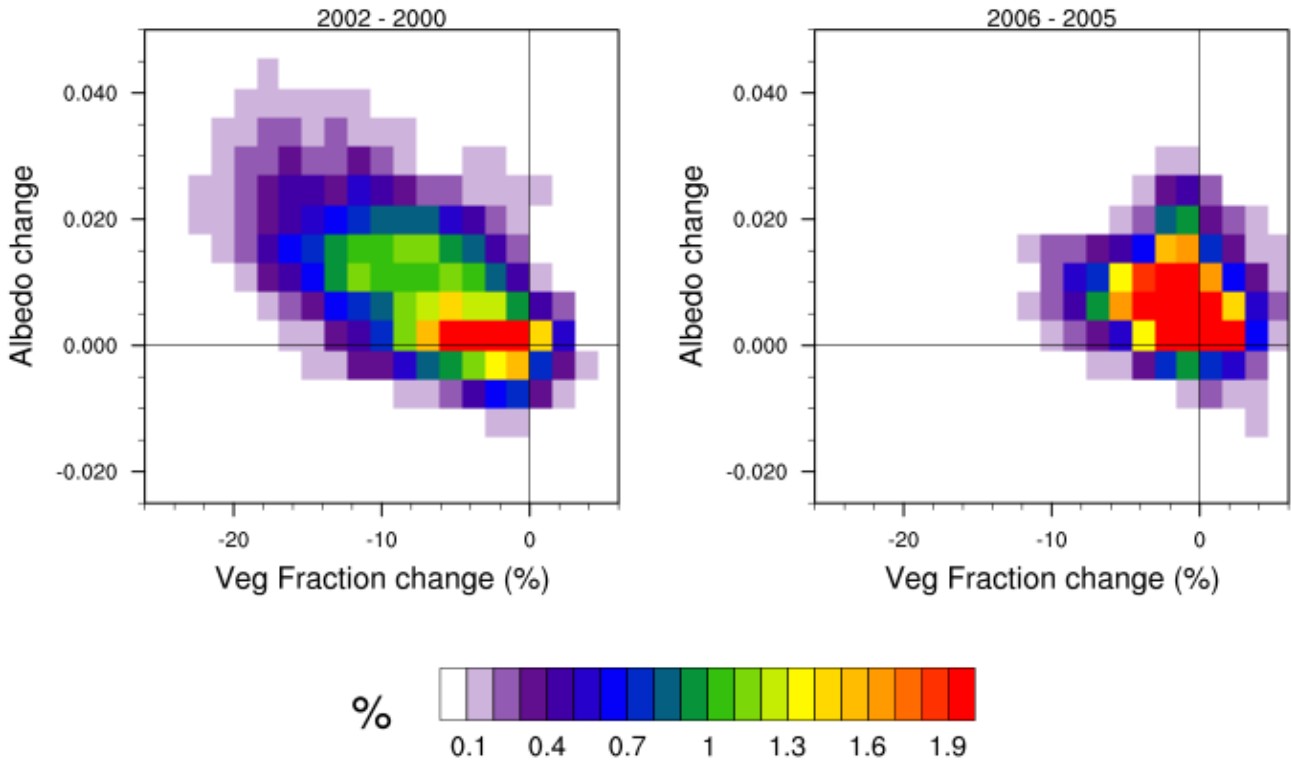

**Figure 9: The bivariate joint probability distribution of the albedo and vegetation changes in the 2002 drought (2002 – 2000) and the 2006 drought (2006-2005). Colours show the percentage of grid cells that fall within each change in albedo/change in vegetation fraction box.**

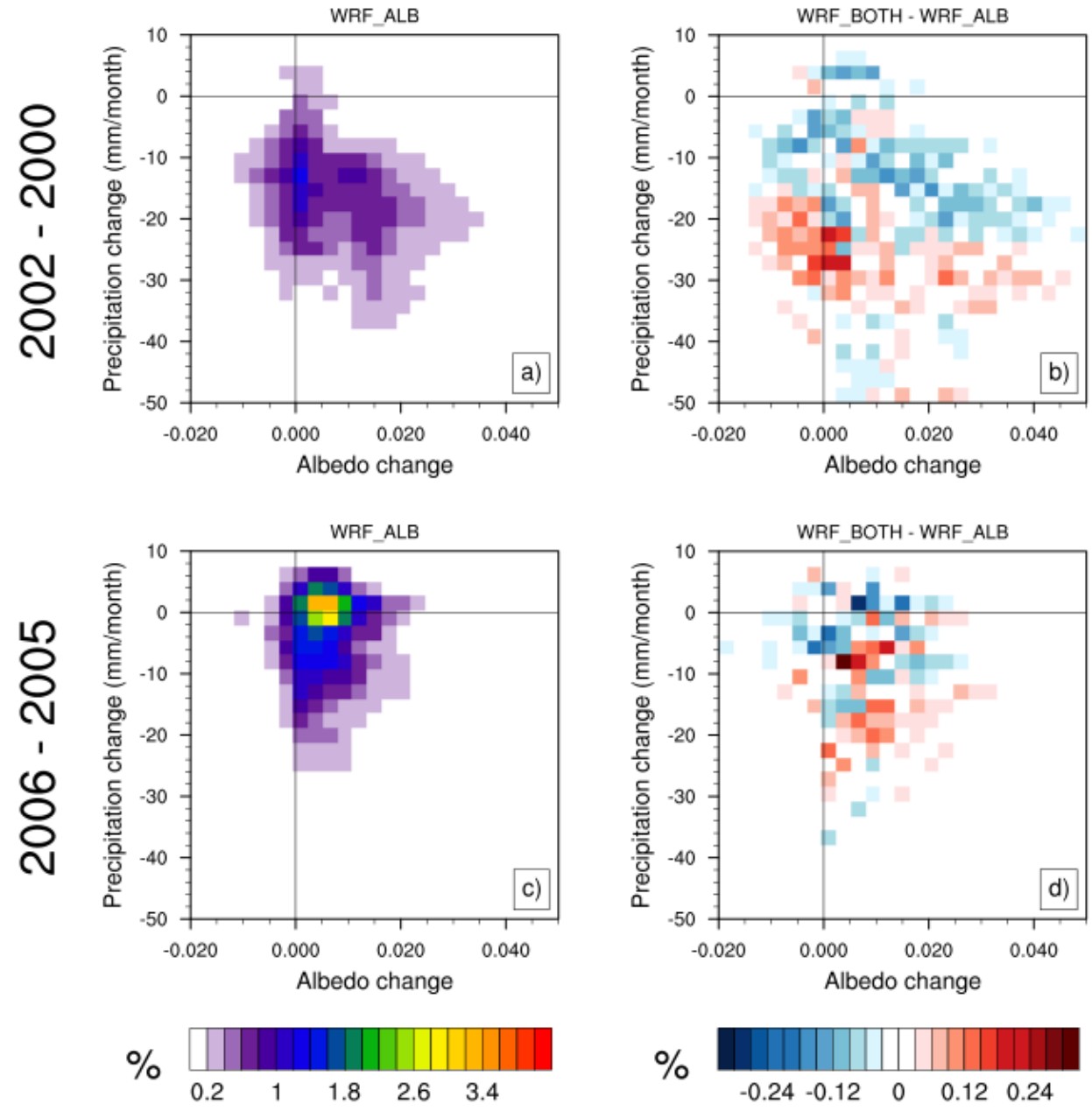

**Figure 10: The bivariate joint probability distribution of the albedo change and precipitation change in the a) 2002 drought (2002 – 2000) and c) the 2006 drought (2006-2005). The change to this relationship caused by the addition of vegetation changes in the WRF_BOTH experiment is shown in b) and d) respectively.**

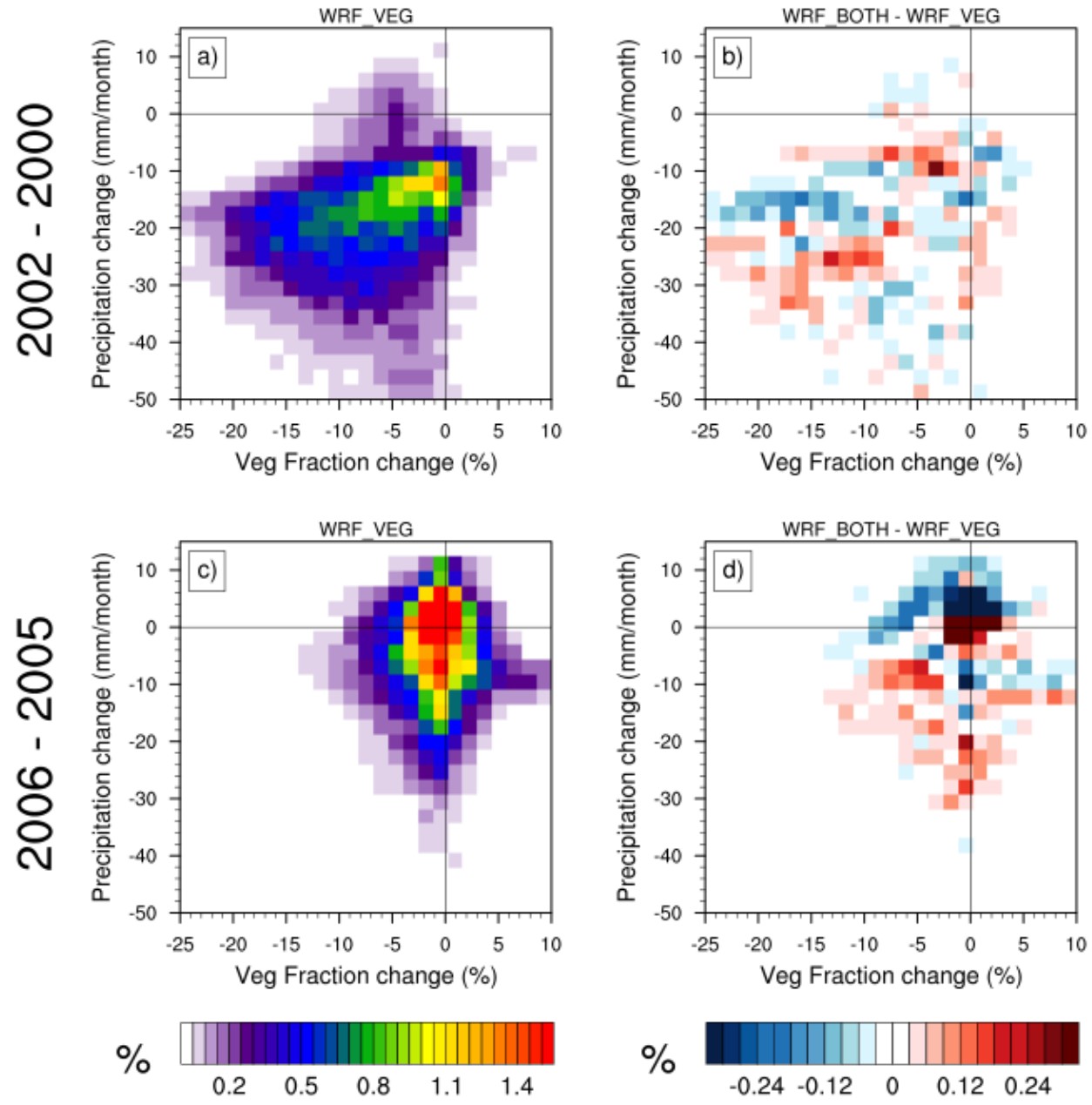

**Figure 11: The bivariate joint probability distribution of the vegetation change and precipitation change in the a) 2002 drought (2002 – 2000) and c) the 2006 drought (2006-2005). The change to this relationship caused by the addition of vegetation changes in the WRF_BOTH experiment is shown in b) and d) respectively.**

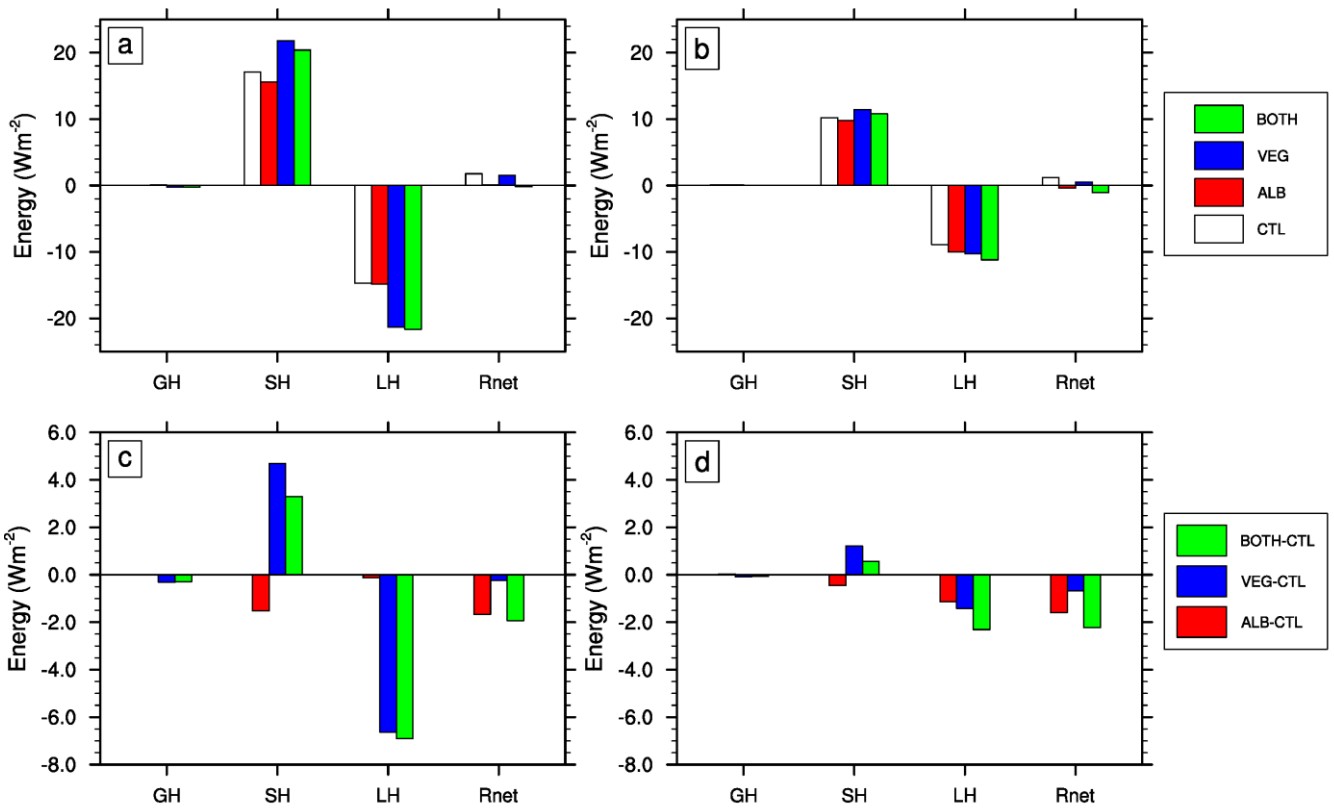

**Figure 12: The change in surface energy budget terms (ground heat (GH), sensible heat (SH), latent heat (LH), net radiation (Rnet)) between drought and pre-drought years. Panels a and b show the 2002-2000 change and 2006-2005 change respectively for each simulation experiment. Panels c and d show the difference between each experiment and the control simulation for the 2002-2000 change and the 2006-2005 change respectively.**

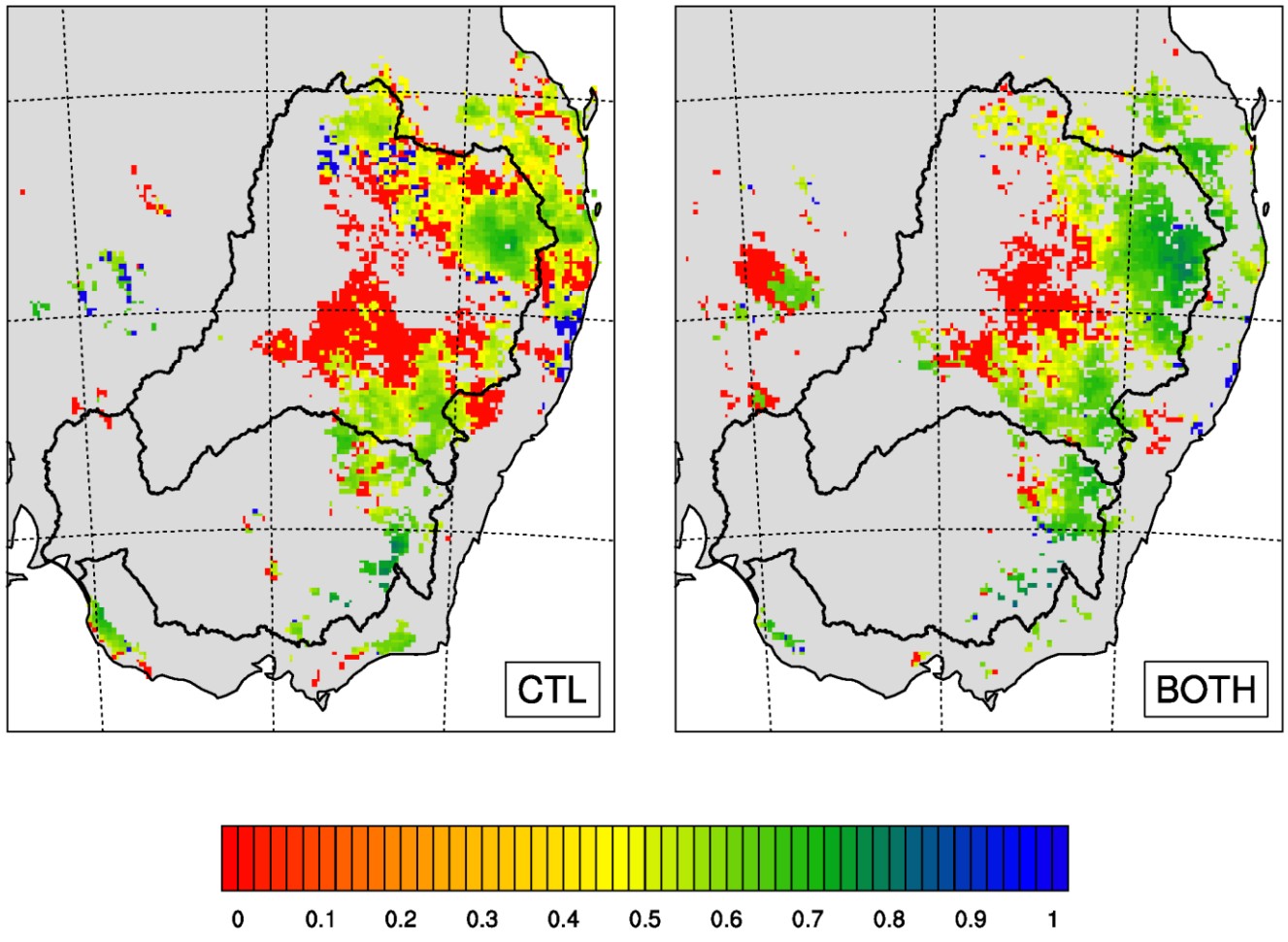

**Figure 13: Ratio of the contribution of decreases in total turbulent fluxes and increases in planetary boundary layer (PBL) height to decreases in moist static energy density in the PBL. Red indicates that only decreases in turbulent fluxes contributes, blue indicates that only increases in PBL height contributes.**

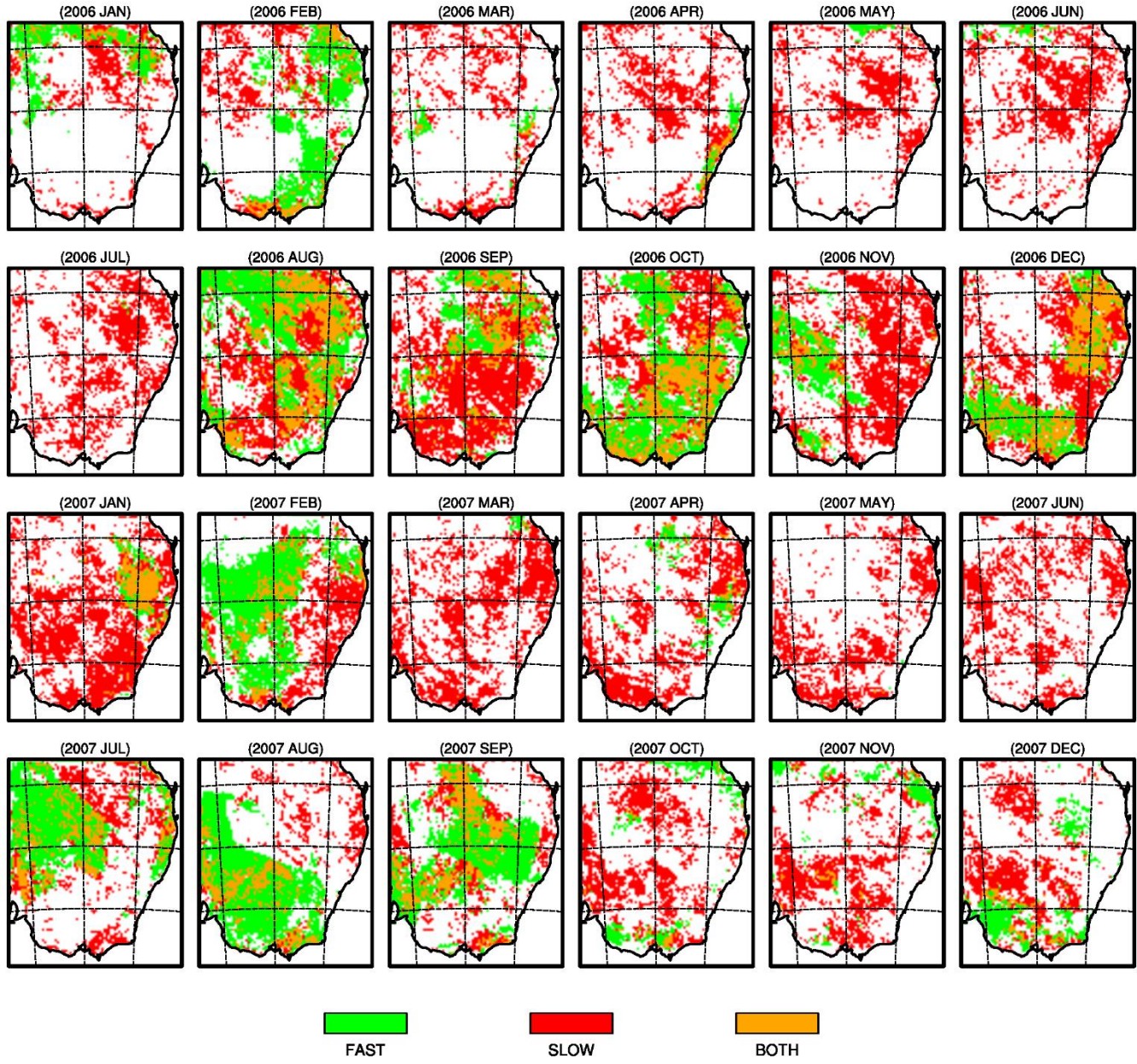

**Figure 14: Distribution of the physical (fast) and biological (slow) mechanisms that exist in monthly and annual variations in the WRF_BOTH simulation in 2006 and 2007.**