# Peer review of "Land surface albedo and vegetation feedbacks enhanced the Millennium drought in south-east Australia"

_Hydrology and Earth System Sciences, 2016_

## Referee Comment (RC1) · Dr. Teuling (Referee) · 19 Sep 2016

The manuscript by Evans et al. addresses the impact of dynamics in vegetation and albedo during the Millennium drought in Australia on simulated climate and precipitation. This is a highly relevant topic, since there are still considerable uncertainties in the contribution of land surface-atmosphere coupling to drought amplification at (sub-)continental scales. The study consists of a relatively straightforward model experiment, and is generally well presented. I have one general comment related to the apparent absence of any process understanding or a priori hypothesis on the role of vegetation and albedo, and some smaller comments that will need to be addressed before possible publication.

**1   Main comment**

My main comment concerns the lack of process understanding. First, no hypothesis is formulated on how and why albedo and vegetation are expected to impact atmospheric conditions. Such a discussion on the role of albedo and vegetation in land surface-atmosphere exchange in semi-arid regions should be added. Secondly, the authors refer to "changes in albedo lead to..." etc., without specifying the direction of the change. This makes it complex for the reader to follow the argumentation, because several cross-checks between the Introduction and Figures and needed to know whether this is consistent with previous studies. This should be improved. Also, the authors did not look into any additional information that can be output by the model, such as soil moisture, temperature or information of the surface energy balance partitioning. An additional figure with analysis of some of these variables could strengthen the story considerably.

**2   Smaller comments**

Albedo anomalies in Figure are plotted with respect to a long-term mean (i.e. not a mean seasonal cycle). I would prefer to see the anomalies with respect to their mean seasonal cycle, so that longer scale deviations are more easily recognized.

A description of the selected land surface parameterization in WRF is lacking. As a result, it is unclear how vegetation fraction affects land surface processes. What is the parameterization/equation(s) were vegetation fraction is used? And how does it affect the evolution of soil moisture?

The authors compare default vegetation and albedo fields from AVHRR with MODIS estimates. These fields differ not only in their inter- and intra-annual variability, but likely also in their mean (at least judging from Figure 3 it seems that most observed
fields have a bias). How much of the precipitation signal can be attributed to the bias rather than interannual variability in vegetation and albedo?

Another comment related to the magnitude of the precipitation signal is whether or not the changes in precipitation are significant when compared to making other more subjective changes in the model, such as choosing different physics packages. In a recent study on heatwave conditions in Europe, Stegehuis et al. (2015) for instance found that the choice for different physics packages significantly affected temperatures and precipitation. This should be discussed.

Concerning the references, a key study on effects of land surface-atmosphere feedbacks on multi-year droughts that seems to be missing is a study by Schubert et al. (2004) on the U.S. Dust Bowl.

**3 References**

Schubert, S. D. et al. (2004). On the cause of the 1930s Dust Bowl. *Science* **303**, 1855–1859.
Stegehuis, A. I. et al. (2015), An observation-constrained multi-physics WRF ensemble for simulating European mega heat waves. *Geosci. Model Dev.*, **8**, 2285–2298.

---

## Referee Comment (RC2) · Anonymous Referee #2 · 26 Sep 2016

Summary: The authors present an RCM study of the Millennium Drought that builds logically from their previous work on the subject. Here they examine interactions between albedo and green vegetation fraction in RCM simulations that include the two peaks of the drought, 2002 and 2006. The paper is well written and both experimental design and results are presented in a clear and logical manner. I believe that the manuscript is appropriate for publication in HESS.

However, I do have two major comments that I would like to see addressed prior to final publication:

1. Methodology: I recognize that AVHRR is a default option for albedo and vegetation fraction in WRF. But the authors clearly have the capability to replace these defaults

with MODIS data, since that is how they are performing their variable albedo and GVF simulations. That being the case, wouldn't the appropriate climatological comparator be a simulation that uses MODIS climatology for these fields? Using AVHRR for the baseline experiments means that differences between the simulations could stem either from interannual variability in MODIS or from differences between the AVHRR and MODIS datasets. Time mean statistics like those presented in Table 1 seem likely to represent changes in dataset rather than the introduction of interannual variability, which is confusing since variability is the topic of the paper. I do not insist that the authors perform new simulations at this point, as the results are sufficiently interesting as they stand. But I would like to understand the choice of design, and I would encourage the authors to distinguish between dataset effects and variability effects throughout their Results and Discussion sections.

2. Mechanism: The manuscript is light on mechanistic interpretation, though members of the authorship team have numerous publications that explore mechanisms associated with these kinds of drought feedbacks. The manuscript would be stronger if it contained a distinct section within the Discussion that addressed mechanism. This would likely require the addition of some results figures or tables, since the manuscript currently doesn't include any results on the surface energy balance or PBL conditions. But I'd expect that such analysis could be added without too much difficulty and could, potentially, substantially elevate interest in this paper.

Minor comments:

1. Abstract: the statement "These results suggest that in terms of drought development, capturing the feedbacks related to vegetation and albedo changes may be as important as capturing the soil moisture-precipitation feedback" seems to come out of nowhere, since the magnitude of the soil moisture feedback has not been mentioned yet. I suggest rephrasing or removing this sentence.

2. Section 3: What meteorological data were used for the offline spinup?

3. Figures 5 & 6: The authors emphasize the fact that WRF_ALB and, in particular, WRF_BOTH show greater differences from WRF_CTL in 2002 than 2000 and in 2006 than 2005. This is clearly supported by the results shown in these figures. But the simulations actually show the greatest difference in 2007. Why was this? Does it fit into a delayed response hypotheses and, if so, does this make the year-after-a-drought response as or more interesting than the results for the drought year? It's also notable that WRF_ALB is low in 2000 and 2001 as well as in 2002. This is not discussed in the text.

---

## Author Comment (AC1) · 18 Nov 2016

Thanks for the thoughtful comments from both reviewers. Our responses are given below in italics.

**Referee #1**
**1 Main comment**
My main comment concerns the lack of process understanding. First, no hypothesis is formulated on how and why albedo and vegetation are expected to impact atmospheric conditions. Such a discussion on the role of albedo and vegetation in land surface-atmosphere exchange in semi-arid regions should be added.

*A paragraph will be added to the introduction to discuss this.*
*"In general terms, mechanisms that produce soil moisture-precipitation feedback involve a change in energy partitioning at the surface that subsequently changes the evolution of the Planetary Boundary Layer (PBL) and the likelihood of triggering precipitation. That is, a decrease in soil moisture leads to more energy being used for sensible heating, an increase in the PBL height with a related decrease in the moist static energy density, resulting in a decreased likelihood of triggering precipitation and a further reduction in soil moisture. Changes in other surface characteristics, such as albedo and vegetation cover, can also change the surface energy partitioning, and produce a similar chain of affects that result in a feedback on the soil moisture conditions. Unlike soil moisture-precipitation feedbacks however, the relationship between changes in these other surface characteristics and the surface energy partitioning can be quite complex. For example, an increase in albedo may reduce the net radiation at the surface, but how will this reduction in available energy be partitioned between the surface fluxes? Or alternatively, a reduction in vegetation cover will mean more exposed soil from which water can evaporate more quickly following a rain storm, and also trigger a reduction in the vegetated area that can continue transpiring through a dry spell (up to a point). That is, vegetation changes have a time varying impact on surface energy partitioning. But what is the cumulative impact on the surface energy fluxes? In reality, soil moisture, albedo and vegetation all change simultaneously. Here we explore the impact of these changes on the development of drought in south-east Australia."*

Secondly, the authors refer to "changes in albedo lead to..." etc., without specifying the direction of the change. This makes it complex for the reader to follow the argumentation, because several cross-checks between the Introduction and Figures and needed to know whether this is consistent with previous studies. This should be improved.

*The text can be altered to include the direction of change where appropriate.*

Also, the authors did not look into any additional information that can be output by the model, such as soil moisture, temperature or information of the surface energy balance partitioning. An additional figure with analysis of some of these variables could strengthen the story considerably.

*Additional analysis can be added to the paper.*
*"4.3 Feedback mechanisms*
*Here we examine the surface energy budget and potential feedback mechanisms during the development of both droughts. In the Murray basin, drought years have less latent heat and more sensible heat as expected (Figure 12a and b). The effect of allowing albedo and vegetation fraction to vary as observed is shown more clearly in Figure 12c and d, which shows the difference in these changes during drought development in each experiment compared to the CTL. For the 2002 drought (Figure 12c), when only the observed albedo increase is included, there is a decrease in surface net radiation (Rnet) compared to CTL. This decrease is used entirely to decrease the sensible heating (SH). This is typical of a water limited environment where water availability*

*controls the latent heat (LH) not the energy. In the 2006 drought the decrease in Rnet is split between LH and SH. Like the 2002 case, these changes are controlled by the water availability. The higher albedo produces a negative feedback on precipitation that is persistent over many years (Figure 6) and results in lower root zone soil moisture, hence less water available for evapotranspiration, compared to CTL.*

[Figure]

Figure 12: The change in surface energy budget terms (ground heat (GH), sensible heat (SH), latent heat (LH), net radiation (Rnet)) between drought and pre-drought years. Panels a and b show the 2002-2000 change and 2006-2005 change respectively for each simulation experiment. Panels c and d show the difference between each experiment and the control simulation for the 2002-2000 change and the 2006-2005 change respectively.

*In the 2002 drought, when only the observed vegetation fraction decrease is included, there is a large decrease in latent heating and a somewhat compensating increase in sensible heating. In this case the vegetation fraction starts higher than the CTL and more transpiration of water from the root zone occurs. During the drought year the vegetation fraction has reduced and the soil moisture depleted such that similar LH occurs in both the VEG and CTL cases. Hence the decrease during drought development is greater in the VEG case. In the 2006 drought a similar but damped response occurs since the available soil moisture is lower in 2005 than 2000.*

*When both the observed albedo increases and vegetation fraction decreases are included, the surface energy balance response is a non-linear combination of the two previous cases. It is worth noting that while the Rnet decrease is similar during the development of the 2002 and 2006 droughts, the change in energy partitioning compared to CTL is three times larger in 2002 than 2006. The lower available soil moisture before the 2006 drought onset has dampened the surface flux changes.*

*These dampened fluxes lead to different feedbacks operating during the development of each drought. Using the methodology in Meng et al. (2014b) to examine the relative contribution to decreases in the moist static energy density of the PBL, we see that the 2006 drought (Figure 13) has a smaller total area with the feedback operating, and the dominant cause is a decrease in the total turbulent heat flux. This differs from the 2002 drought (figure 14 in Meng et al., 2014b) where the dominant cause is an increase in the PBL height. These feedbacks act relatively quickly and are mostly related to changes in albedo and the current soil moisture state.*

[Figure]

Figure 13: Ratio of the contribution of decreases in total turbulent fluxes and increases in planetary boundary layer (PBL) height to decreases in moist static energy density in the PBL. Red indicates that only decreases in turbulent fluxes contributes, blue indicates that only increases in PBL height contributes.

*To account for the different time scales associated with albedo and vegetation changes, the methodology in Meng et al. (2014a) can be used to identify the presence of fast physical feedbacks associated with albedo and soil moisture changes as discussed above, and slower vegetation related changes that impact the strength of the fast feedbacks. Figure 14 shows when the fast physical and slow biological mechanisms are active during the 2006 drought and is comparable to figure 12 in Meng et al. (2014a), which shows the same thing for the 2002 drought. The findings confirm that the dampened surface fluxes during the development of the 2006 drought result in less area exhibiting the feedbacks compared to 2002, particularly the slow biological feedback. It also concurs with the finding in Meng et al. (2014a) that the fast feedback is less likely to occur if the*

*slow feedback is present (relatively few orange areas), as it acts to reduce the soil moisture changes.*

[Figure]

Figure 14: Distribution of the physical (fast) and biological (slow) mechanisms that exist in monthly and annual variations in the WRF_BOTH simulation in 2006 and 2007."

**2  Smaller comments**
Albedo anomalies in Figure are plotted with respect to a long-term mean (i.e. not a mean seasonal cycle). I would prefer to see the anomalies with respect to their mean seasonal cycle, so that longer scale deviations are more easily recognized.

*The figure is plotted to include the seasonal cycle as some substantial differences in the seasonal cycle of the default and observed vegetation fraction exist. These seasonal differences are more difficult to interpret using anomalies from the mean seasonal cycle.*

A description of the selected land surface parameterization in WRF is lacking. As a result, it is

unclear how vegetation fraction affects land surface processes. What is the parameterization/equation(s) were vegetation fraction is used? And how does it affect the evolution of soil moisture?

*An extra paragraph can be added to section three that discusses the role of vegetation fraction and albedo as parameterised within the land surface model.*
*"The Noah land surface scheme is described in Chen and Dudhia (2001). In this implementation the green vegetation fraction is used to determine the fraction of a grid cell that is covered by vegetation vs bare soil. It has a direct impact on the partitioning of evaporation between soil evaporation, canopy evaporation and transpiration. The albedo changes the amount of upward shortwave radiation and hence the energy available for use in driving surface energy fluxes."*

The authors compare default vegetation and albedo fields from AVHRR with MODIS estimates. These fields differ not only in their inter- and intra-annual variability, but likely also in their mean (at least judging from Figure 3 it seems that most observed fields have a bias). How much of the precipitation signal can be attributed to the bias rather than interannual variability in vegetation and albedo?

*The initial simulation evaluation is done over the 8 year period. The bias shown in Table 1 and Figure 4 largely reflects the bias between driving datasets. While the other statistics reported are also affected by the intra- and inter-annual variability. A paragraph can be added on this point.*
*"Note that two factors are being tested here. First, the default albedo and vegetation fraction datasets represent climatological conditions in the late 1980s and not during the time of interest. Substantial changes in the land surface may have occurred over the intervening 20 years. There may also be some offset between the AVHRR (default) and MODIS (observed) sensors. Most of the bias between the default and observed datasets may be due to this temporal and sensor mismatch. Second, the default datasets do not capture the inter-annual variability associated with drought development. This mismatch between the default and observed datasets will have some impact on the RMSE and pattern correlation statistics. The effect of this inter-annual variability is the focus of sections 4.2 and 4.3, which examine changes in time within each simulation. Thus the influence of between simulation biases are largely removed."*

Another comment related to the magnitude of the precipitation signal is whether or not the changes in precipitation are significant when compared to making other more subjective changes in the model, such as choosing different physics packages. In a recent study on heatwave conditions in Europe, Stegehuis et al. (2015) for instance found that the choice for different physics packages significantly affected temperatures and precipitation. This should be discussed.

*Indeed several studies have shown that land-atmosphere feedbacks are model dependant. A limitation of this study is the use of a single model configuration. A caveat concerning this point could be added to the end of the discussion.*

Concerning the references, a key study on effects of land surface-atmosphere feed-backs on multi-year droughts that seems to be missing is a study by Schubert et al. (2004) on the U.S. Dust Bowl.

*Reference can be added in the introduction.*

**Referee # 2**
1. Methodology: I recognize that AVHRR is a default option for albedo and vegetation fraction in WRF. But the authors clearly have the capability to replace these defaults with MODIS data, since

that is how they are performing their variable albedo and GVF simulations. That being the case, wouldn't the appropriate climatological comparator be a simulation that uses MODIS climatology for these fields? Using AVHRR for the baseline experiments means that differences between the simulations could stem either from interannual variability in MODIS or from differences between the AVHRR and MODIS datasets. Time mean statistics like those presented in Table 1 seem likely to represent changes in dataset rather than the introduction of interannual variability, which is confusing since variability is the topic of the paper. I do not insist that the authors perform new simulations at this point, as the results are sufficiently interesting as they stand. But I would like to understand the choice of design, and I would encourage the authors to distinguish between dataset effects and variability effects throughout their Results and Discussion sections.

*This is a good point and deserves further explanation in the manuscript. Indeed two (not independent) factors are tested in this experiment.*
*1. The impact of using the default albedo and vegetation fraction datasets that represent climatological conditions in the late 1980s instead of the observed conditions at the time of interest (20 years later) – as is done in almost all studies with this model.*
*2. The impact of inter-annual variability on the evolution of drought.*
*Currently the first is addressed in section 4.1, while the second is addressed in section 4.2.*
*Text could be added to the manuscript to clarify this choice of design*
*"Note that two factors are being tested here. First, the default albedo and vegetation fraction datasets represent climatological conditions in the late 1980s and not at the time of interest. Substantial changes in the land surface may have occurred over the intervening 20 years. There may also be some offset between the AVHRR (default) and MODIS (observed) sensors. Most of the bias between the default and observed datasets may be due to this temporal and sensor mismatch. Second, the default datasets do not capture the inter-annual variability associated with drought development. This mismatch between the default and observed datasets will have some impact on the RMSE and pattern correlation statistics. The effect of this inter-annual variability is the focus of sections 4.2 and 4.3, which examine changes in time within each simulation. Thus the influence of between simulation biases are largely removed."*

2. Mechanism: The manuscript is light on mechanistic interpretation, though members of the authorship team have numerous publications that explore mechanisms associated with these kinds of drought feedbacks. The manuscript would be stronger if it contained a distinct section within the Discussion that addressed mechanism. This would likely require the addition of some results figures or tables, since the manuscript currently doesn't include any results on the surface energy balance or PBL conditions. But I'd expect that such analysis could be added without too much difficulty and could, potentially, substantially elevate interest in this paper.

*Additional analysis of the energy balance and feedbacks to the PBL and precipitation will be added to the discussion. See a more thorough response in the answer to referee 1 above.*

**Minor comments:**
1. Abstract: the statement "These results suggest that in terms of drought development, capturing the feedbacks related to vegetation and albedo changes may be as important as capturing the soil moisture-precipitation feedback" seems to come out of nowhere, since the magnitude of the soil moisture feedback has not been mentioned yet. I suggest rephrasing or removing this sentence.

*The sentence could be rephrase as "These results, when compared with previous studies of land-surface feedbacks and drought, suggest that capturing the feedbacks related to vegetation and albedo changes may be as important as capturing the soil moisture-precipitation feedback for drought development."*

2. Section 3: What meteorological data were used for the offline spinup?

*Offline spinup was not performed. 15 years of online coupled spinup was performed using boundary conditions from the NCEP-NCAR reanalysis (3rd paragraph of section 3).*

3. Figures 5 & 6: The authors emphasize the fact that WRF_ALB and, in particular, WRF_BOTH show greater differences from WRF_CTL in 2002 than 2000 and in 2006 than 2005. This is clearly supported by the results shown in these figures. But the simulations actually show the greatest difference in 2007. Why was this? Does it fit into a delayed response hypotheses and, if so, does this make the year-after-a-drought response as or more interesting than the results for the drought year? It's also notable that WRF_ALB is low in 2000 and 2001 as well as in 2002. This is not discussed in the text.

*The year 2007 does show the largest differences between WRF_CTL and WRF_ALB or WRF_BOTH in Figure 6. Figure 5 shows that this is a period of recovery from drought and hence is not a focus of this paper (which is investigating drought development). It does however indicate that the observed albedo changes cause a delay in the drought recovery. Text to this affect could be added to the manuscript. The fact that WRF_ALB is low in 2000 and 2001 (indeed, is lower than WRF_CTL almost always) is related to the albedo dataset change, which results in a small but persistent decrease in precipitation compared to WRF_CTL (seen in Figure 6) while the drought evolution is very similar (Figure 5).*